# Security Awareness in Smart Homes and Internet of Things Networks through Swarm-Based Cybersecurity Penetration Testing

**Thomas Schiller [1,\*], Bruce Caulkins [2], Annie S. Wu [3] and Sean Mondesire [1,\*]**

[1] School of Modeling Simulation and Training, University of Central Florida, Orlando, FL 32816, USA
[2] Institute of Simulation and Training, University of Central Florida, Orlando, FL 32816, USA
[3] Department of Computer Science, University of Central Florida, Orlando, FL 32816, USA
[\*] Correspondence: th258533@ucf.edu (T.S.); sean.mondesire@ucf.edu (S.M.)

**Abstract:** Internet of Things (IoT) devices are common in today's computer networks. These devices can be computationally powerful, yet prone to cybersecurity exploitation. To remedy these growing security weaknesses, this work proposes a new artificial intelligence method that makes these IoT networks safer through the use of autonomous, swarm-based cybersecurity penetration testing. In this work, the introduced Particle Swarm Optimization (PSO) penetration testing technique is compared against traditional linear and queue-based approaches to find vulnerabilities in smart homes and IoT networks. To evaluate the effectiveness of the PSO approach, a network simulator is used to simulate smart home networks of two scales: a small, home network and a large, commercial-sized network. These experiments demonstrate that the swarm-based algorithms detect vulnerabilities significantly faster than the linear algorithms. The presented findings support the case that autonomous and swarm-based penetration testing in a network could be used to render more secure IoT networks in the future. This approach can affect private households with smart home networks, settings within the Industrial Internet of Things (IIoT), and military environments.

**Keywords:** cybersecurity; penetration testing; IoT; swarm; smart home

## 1. Introduction

Computer networks today inherit devices commonly known as Internet of Things (IoT) devices. IoT devices are characterized as objects that are connected to the internet [1]. These devices are present in smart homes where they can be found as smart TVs, smart fridges, or smart speakers. An estimate of 11 devices per person per household can be found of these devices today, with the prognosis of a rise to more than 30 per person per household by 2030 [2]. These devices are very powerful and often run fully functional operating systems like Linux [3].

IoT devices can further be found in industrial environments as controllers and machines connected to the internet [4]. They are even becoming increasingly important in today's military applications, where they are represented as sensors or vehicles of different forms [5].

IoT devices can provide benefits in these environments. An example of smart homes could be household appliances that can be started remotely and use the power grid when energy is cheap or when lots of renewable energy is produced. These devices can also inform users when tasks are finished or problems occur. In industrial settings, controllers can communicate with each other to limit times of delay and provide higher productivity. Military settings can benefit, for example, from a higher density of sensors that share information and aggregate information.

However, networks of IoT devices are prone to cybersecurity attacks. Cybersecurity penetration testing is a common method for active cybersecurity enhancement to detect vulnerabilities before the breach.

This study identified two research gaps regarding the cybersecurity of IoT networks: First, no literature was found on autonomous penetration testing using multiple agents and swarm intelligence. Second, no literature was found on using IoT devices for penetration testing other IoT devices.

A standard method with a single penetration tester can detect vulnerabilities in a computer network, but this process can be slow and may not identify every vulnerability. However, multiple penetration testers processing the testing in parallel instead of one tester processing the testing serially has both benefits and downsides. For instance, numerous testers may have a higher detection rate than a single tester since multiple agents execute all actions simultaneously. Even so, this can result in duplicated work and wasted resources because of missing communication between agents. This work addresses this lack of communication by applying swarm-based intelligence to coordinate multiple agents as they perform parallel penetration testing.

This work is driven by the research question: To what extent does swarm-based IoT network penetration testing detect active vulnerabilities beyond linear and sequential penetration testing?

The novelty and contribution of this work are to provide a method of using the capabilities of IoT devices to create more secure smart homes and IoT networks with the utilization of swarm intelligence and autonomous cybersecurity penetration testing. Although this work addresses smart home scenarios, the impact of this work can be extended to industrial and military settings with future research.

## 2. Related Work

The exploitable IoT, cybersecurity measures, and autonomous penetration testing all play a part in modern public and private networks. The Exploitable IoT section discusses work on the possible dangers of IoT devices. The Cybersecurity Measures section explains how to address possible vulnerabilities in these devices. Lastly, the Autonomous Penetration Testing section explores how security measures can be developed with humans out of the loop. This section ends with the presented work's research gaps, problem statement, and research question.

### 2.1. Exploitable IoT

The capabilities of IoT devices, like connectivity and computational power, render them prone to cybersecurity issues. These devices can face several vulnerabilities, such as inadequate physical security, unnecessary open ports, or insufficient access control [6]. Consequently, IoT devices in any context can become targets for cybercrime and impact consumers and industrial applications. This can cause significant harm regarding data confidentiality, integrity, and availability, which are essential for a connected society that shares sensitive information over the internet. Furthermore, it can also lead to physical harm. One example of IoT exploitation is the remote hack of the Controller Area Network (*CAN*) bus of a 2014 Jeep Cherokee, in which an ethical hacker was able to take control of the driving subsystem, including steering and braking [7]. In 2016, St. Jude Medical, Inc. (Saint Paul, MN, U.S.A.) had to announce that their pacemakers needed to be recalled and remediated due to cybersecurity risks. It was possible to remotely access these pacemakers and change rates or initiate a battery drain attack, which could be medically harmful to users [8]. Additionally, in 2016, the Mirai botnet infected more than 500,000 IoT devices worldwide to form one of the largest botnets globally. A botnet is a network of internet-connected devices that runs malicious services. Mirai infected IoT devices such as digital video recorders, routers, or surveillance cameras. These devices were then used to execute DDoS attacks in exchange for a fee [9]. Detecting vulnerabilities, malicious behavior, and exploits is crucial and is the focus of the field of cybersecurity.

More recent literature shows that IoT devices are often battery powered and use the Low-Power Wide-Area Network (LPWAN). The LPWAN does come with security constraints, especially when protocols like Long-Range Wide Area Networks (LoRaWAN) are used [10].

### 2.2. Cybersecurity Measures

Detecting vulnerabilities, malicious behavior, and exploits falls in the field of cybersecurity. Kaur et. al. provide a comprehensive definition of cybersecurity: "Cybersecurity refers to designing, developing, and using technologies, processes, and practices to protect organizational assets, customer data, and intellectual property from intentional or unintentional breaches by unauthorized personnel." [11] (p. 17). The techniques, technologies, and processes used in cybersecurity are diverse and critical in securing network environments. One technique involves detecting network vulnerabilities and suspicious behavior, which can be done *passively* or *actively*.

*Passive detection* includes activities that scan network or data traffic and detect malicious traffic. Examples of this include firewalls or anti-virus software. Much research has been conducted in this field, resulting in methods to make detection automated or even autonomous. Studies have shown that machine learning (*ML*) can be applied to detect malicious network behavior with high accuracy. Excellent results were demonstrated with unsupervised ML for differentiating botnet traffic from normal network traffic, with 94.78% accuracy on a UNBS-NB 15 dataset and 98.08% on a KDD99 dataset [12]. In 2021, high accuracy and precision were achieved with ML classifiers using datasets collected via honeypots [13]. Moreover, in 2021, *Artificial Neural Network Particle Swarm Optimization* (*ANN PSO*) was used to apply ML and swarm optimization to botnet detection with high accuracy [14]. Despite the high accuracy and precision in botnet detection achieved in these works, there are limitations. These methods can only detect an attack after it has already occurred or while it is performed. Once the cybercriminal discovers a vulnerability, it may be too late to prevent exploitation.

*Active methods* are available for detecting vulnerabilities and cybersecurity deficiencies before a breach and before any action by black hat hackers. One of these active methods is penetration testing, also known as ethical hacking. Penetration testing is a structured process for testing networks, systems, organizations, and employees for security vulnerabilities. According to Shebli and Beheshti, "A penetration test is used to identify the risks that may occur when an attacker gets access to the organization's computing system and networks. Performing a PEN test [penetration test] will help estimate the mitigation plan to close security gaps before the actual attack happens" [15].

There are three different penetration testing methods: black box, grey box, and white box penetration testing. With black box penetration testing, the tester knows nothing about the network; with grey box testing, some information is provided; and with white box penetration testing, the tester knows detailed internal information about the network and system being tested. A white box test would correspond to an attack by an employee with internal knowledge of networks and security measures. Penetration testing consists of different phases. The execution phases are information gathering, vulnerability analysis, and vulnerability exploits. The test preparation and the test analysis phases introduce and conclude the other phases [15].

Penetration testing can be performed using various available tools that allow for scanning networks, enumerating information, and performing exploits. A common operating system used for penetration testing is Kali Linux, which includes tools for reconnaissance, scanning, enumeration, and exploitation [16]. Epling et al. demonstrate that the microcomputer Raspberry Pi B+ could be used as a penetration testing device [17]. Since then, the computational power of these microcomputers has increased significantly, with the Raspberry Pi version 4b being much faster than version B+ [18].

IoT devices can be penetration tested, and recent literature has demonstrated how to perform this to detect vulnerabilities in smart refrigerators running Tizen Linux. These

devices were secure against DDoS or DNS attacks but used unencrypted communication and allowed simple passwords [19,20]. Ethical hacking on smart televisions using Tizen Linux in 2021 resulted in the possibility of root-shell execution, even though Tizen's security mechanisms were active [21].

Penetration testing relies heavily on human effort for various reasons: the test preparation phase, involving the planning and creation of the test scope and addressing legal questions, requires diverse fields of human knowledge that are challenging to automate. The technical aspect of penetration testing in the execution phase requires experts and specialists with years of experience in applying and setting up the tools [15,16].

Penetration testing can be performed independently of human involvement in specific areas. This study categorizes these efforts along with a characterization known from the application of artificial intelligence: *Human-in-the-Loop* (*HITL*), *Human-on-the-Loop* (*HOTL*), and *Human-out-of-the-Loop* (*HOOTL*). This human-role model is used, for example, in military applications [22] and self-driving cars [23]. This model is applied to penetration testing as a novelty. HITL describes a concept where the human controller controls the systems in real time. HOTL systems work independently, but the human controller can always interact with the system, and the system refers to the human in critical situations. HOOTL systems are autonomous and require no human interaction.

An example of HITL in penetration testing is an expert who executes a program for scanning the network. In contrast to this traditional penetration testing with human experts executing every step of the penetration testing, options are available to limit the human effort in the process. By utilizing the capabilities of automatic functions of penetration testing tools, it is possible to reduce human interaction. This is an example of HOTL penetration testing. Abu-Dabaseh and Alshammari have shown that many tools used in the test implementation phase can be automated [24]. Tools like The Harvester, Nessus, or Metasploit can run automatically. Logging, auditing, and reporting can also be automated. Although this method can save time and human effort, there is still a limitation in executing these tools and setting the scope before the execution. These steps are heavily reliant on expert knowledge.

It is notable that IoT networks have domain-specific cybersecurity characteristics, such as smart grid networks. Comprehensive security frameworks such as SDN-microSENSE and SPEAR SIEM address these characteristics and provide a wide palette for enhancing IoT network security [25,26].

*2.3. Autonomous Penetration Testing*

HOOTL autonomous penetration testing aims to decrease the human role in the process further but faces the challenge of how to deal with uncertainty. This challenge can be addressed using attack trees or attack graphs, as demonstrated in early approaches where each node of the graph or tree represents a possible attack state [27]. Attack graphs have also served in recent approaches and have been improved in being loop-free to be scalable and useful for large cloud networks with services leaving and entering the cloud frequently [28]. Recent research also employed machine learning algorithms. Kachare et al. show an approach to detect malicious IoT devices by utilizing a sandbox approach and machine learning algorithms which could be applied in an autonomous setting [29]. Recent approaches to autonomous penetration testing often use reinforcement learning (*RL*). RL, a subset of ML, mimics learning processes originating from psychology and B. F. Skinner's radical behaviorism theory [30]. RL trains agents to perform behaviors by giving positive or negative rewards for actions. Over time, agents adjust their actions according to the rewards to optimize outcomes [31]. The ASAP framework displayed a method to create autonomous and non-intuitive attack plans [32]. *Markov Decision Processes* (*MDPs*) and *Partial Observable MDPs* (*POMDPs*) were used to demonstrate how to perform penetration testing, with RL addressing uncertainty [33]. Due to the lack of a penetration testing testbed for RL training, Schwartz created a network simulator. The results indicated that RL could solve penetration testing subproblems by exploiting devices in a simple simulated

network scenario [33]. This network simulator was later used to solve penetration testing subproblems with RL and Deep Q-Networks [34]. The OpenAI Gym environment with its simulator CybORG 2020 was used to address penetration testing subproblems with RL in a simulated environment [35]. OpenAI Gym provides structured and reproducible methods and tools for RL. This work was later extended to use an emulated environment based on Amazon's AWS service [35–37]. In 2020, it was demonstrated that RL and self-play can identify security strategies in a simple attacker–defender game represented in a Markov game [38]. These previous works show that autonomous agent-based approaches and RL can address penetration testing subproblems in the technical phase. However, the research is still in its infancy, with limitations stemming from single-agent approaches focused on penetration testing subproblems with simplified networks.

In 2022, a first multi-agent approach using a multi-agent model was conducted, aiming to attack an attacker node in a network or isolate attacked nodes as the defender [39]. This work's limitation was the simplified network and penetration testing simulation, with each side (red and blue) consisting of only one agent, making it essentially a single-agent penetration testing simulation.

A recent review from April 2023 on RL applications in cyber security displays previous work using multiple RL agents to detect network intrusions or malicious network traffic [40]. However, this work is limited to passive measures like detecting malicious network traffic. No work was displayed using multiple agents for the active measure of penetration testing.

### 2.4. Research Gaps

The literature review identified two research gaps: First, no literature was found on autonomous penetration testing using multiple agents and swarm intelligence. Second, no literature was found on using IoT devices for penetration testing other IoT devices.

This work combines these research gaps into a novel field of research: HOOTL, autonomous penetration testing of IoT devices by IoT devices utilizing multiple agents, and swarm intelligence.

### 2.5. Problem Statement and Research Question

A single penetration tester can detect vulnerabilities in a computer network, but this process can be slow and may not identify every vulnerability. However, multiple penetration testers processing the testing in parallel instead of one tester processing the testing serially have both benefits and downsides. For instance, numerous testers may have a higher detection rate than a single tester because multiple agents execute all actions simultaneously. Even so, this approach may result in duplicated work and wasted resources because of missing communication between agents. The presented work addresses the lack of communication by applying swarm-based intelligence to coordinate multiple agents as they perform parallel penetration testing. Information sharing and task allocation could improve coordination, reduce duplicate work, and improve efficiency in multi-agent penetration testing.

The research gap and problem statement demonstrated in the literature review necessitate the following research question: To what extent does swarm-based IoT network penetration testing detect active vulnerabilities beyond linear and sequential penetration testing?

## 3. Materials and Methods
### 3.1. Hypotheses

The overall detection rate of unique detected vulnerabilities is demonstrated with inferential statistics and tested with the following hypotheses:

**Hypothesis 1.** *Linear multi-agent penetration testing by other IoT devices in the same network has a better rate of detecting unique vulnerabilities than linear single-agent penetration testing.*

**Hypothesis 2.** *Swarm-based penetration testing by other IoT devices in the same network utilizing a queue-based algorithm has a better rate of detecting unique vulnerabilities than linear multi-agent penetration testing.*

**Hypothesis 3.** *Swarm-based penetration testing by other IoT devices in the same network utilizing a nature-based PSO algorithm has a better rate of detecting unique vulnerabilities than linear multi-agent penetration testing.*

**Hypothesis 4.** *Swarm-based penetration testing by other IoT devices in the same network utilizing the PSO-based algorithm has a better rate of detecting unique vulnerabilities than swarm-based penetration testing utilizing the queue-based algorithm.*

Task allocation and information sharing in a swarm can optimize detection rates and improve detection speed. Therefore, descriptive statistics are used to demonstrate whether swarm-based algorithms detect vulnerabilities faster than multiple agents using a linear algorithm.

### 3.2. Research Objectives

Based on the research question and the hypotheses, this work investigates the following research objectives. Research objective 1 is to observe detection rates and the detection speed of vulnerabilities in an IoT network with multi-agent linear penetration testing compared with a single-agent linear approach. Research objective 2 is to observe detection rates and the detection speed of vulnerabilities in an IoT network with swarm-based multi-agent penetration testing compared to a multi-agent linear penetration testing approach. Research objective 3 is to investigate how a nature-based swarm algorithm would compare to a queue-based swarm algorithm regarding the detection rate and detection speed. Research objective 4 is to investigate how swarm algorithms perform on a larger scale in a smart building with more IoT devices and, therefore, more agents in the swarm than with a smart home scale.

### 3.3. Simulation Environment

A custom simulation environment, called *CyberSim-SwarmIoT*, was developed in Python 3 and used for the experiments in this work. This environment is a text-based, multi-agent, constructive network simulator. The simulation core is based on the network simulator CyberSim [41]. It provides the simulation framework with the initialization of the simulation and the execution of each simulation step. In CyberSim-SwarmIoT, multiple agents can act in a network with simplified network activities. Examples of these activities are ping, nmap, netstat, and port scan. Furthermore, agents can perform attacks (e.g., password cracking) on other agents. Permitted activities and the success probabilities of network actions are stored in a look-up table. Agents receive feedback from their actions via a blackboard. Actions from other agents affecting an agent are also written on a blackboard. The blackboard is the only information that the agents acquire from the simulation. There is no global knowledge, but agents can store information locally for reuse in later simulation steps. The simulation is stopped after a certain amount of timesteps or after the entire vulnerability set in the network configuration table is detected by the agents.

### 3.4. Algorithms

Three different algorithms for CyberSim-SwarmIoT were developed for the experiments in this work: one *linear penetration testing algorithm* and two *swarm-based penetration algorithms* to investigate the parallel penetration testing of IoT networks.

#### 3.4.1. Linear Penetration Testing Algorithm

The first and simplest algorithm is a single linear penetration testing algorithm. This algorithm can be used with a single agent or with multiple agents. Figure 1 displays the

algorithm's logic. This single linear penetration testing algorithm mimics human penetration testing behavior in a simplified manner. This human penetration testing behavior is based on the NIST Technical Guide to Information Security Testing and Assessment, Special Publication 800-115 [42]. According to this document, the steps of planning, discovery, attack, and reporting are processed in that order. Due to the simplicity of the simulation, the only steps processed are discovery and attack. Therefore, the algorithm scans, enumerates, and attacks in this order. That is, only after the scanning and enumeration have occurred, the algorithm begins attacking discovered potential vulnerabilities. This algorithm is used by a single agent to observe single linear penetration testing and by multiple agents when conducting linear multi-agent penetration testing.

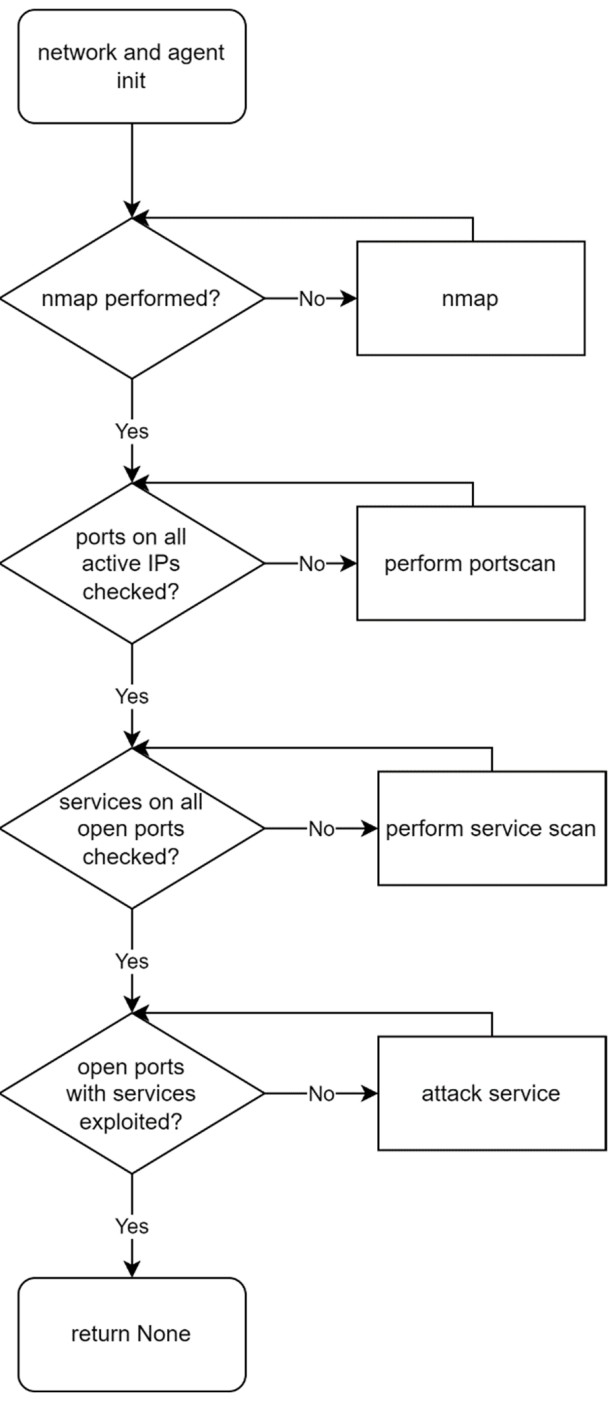

**Figure 1.** Single linear penetration testing algorithm in CyberSim-SwarmIoT.

### 3.4.2. Queue-Based Swarm Penetration Testing Algorithm

The queue-based swarm algorithm (Figure 2) utilizes queues that function as stacks (Last-In-First-Out (LIFO)). The agent's first step is a network scan (nmap) to generate a network table of active devices in the network. In the following steps, the algorithm uses the information stored in the queues to perform actions. After processing one queue entry, the algorithm deletes this chunk of information from the queue. Four distinct queues are used for this algorithm, the first of which stores known IP addresses on the network. The second queue stores IP addresses with ports that can be scanned along with this IP address. Additionally, the third queue stores services that can be scanned, and the fourth queue further stores attack actions that can be performed. Given this structure, the queues contain the following information: queue 1 (Q1), IPs to be scanned; queue 2 (Q2), ports to be scanned; queue 3 (Q3), open ports to be scanned for services; and queue 4 (Q4), attacks to be performed.

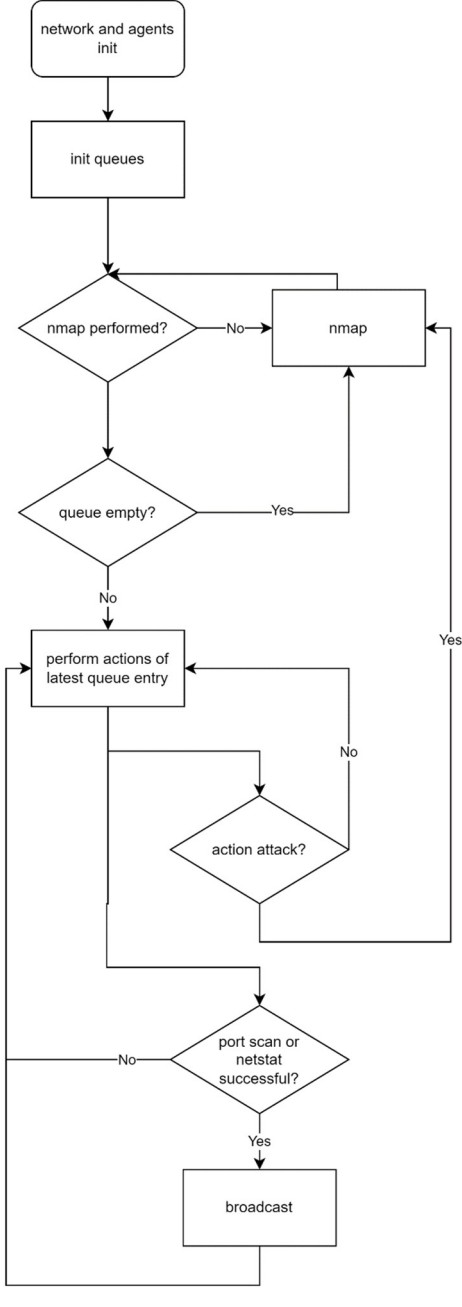

**Figure 2.** Swarm-based penetration testing algorithm utilizing queues in CyberSim-SwarmIoT.

The queues are used by the agent from bottom to top. Therefore, if the agent can perform an action, it first checks queue 4 and uses the first entry when this queue is filled. If a queue is empty, the next queue in reverse order is used. If every queue is empty, a network scan is performed again, producing the following order of action: simulation start, nmap, use Q4, use Q3, use Q2, use Q1, nmap, etc.

The queues are filled according to the information gained by the agent. A known active IP from the network scan will automatically make the second queue fill with every possible IP–port combination for this active IP. Thus, the second queue will be filled after the network scan. A known open port on an active IP will produce IP–port combinations in the third queue that can be scanned for active services with the netstat command. Whenever queues are filled, one checks whether this information has already been in the queue. Therefore, no action will be performed twice by the same agent.

It is in the swarm algorithm's nature for the agents to communicate, which occurs via messages that can be sent from one agent to another. In the queue-based swarm algorithm, the agents communicate when they discover an open port or active service. For this process, the agent enters a communication phase, where it sends messages to other randomly assigned agents. The communication intensity can be defined in the initialization phase of the agent algorithm codebase in CyberSim-SwarmIoT. The higher the number is, the longer the communication takes and the more agents the communication reaches. When one agent receives information from another, it uses this information to fill its queues according to the process described earlier.

### 3.4.3. PSO-Based Swarm Penetration Testing Algorithm

The third algorithm and second swarm algorithm developed and used for this work is nature-based and displayed in Figure 3. This algorithm is based on Particle Swarm Optimization (PSO), which mimics the behavior of a flock of birds and results in some agents following others on a path to an optimum or solution. The algorithm has two states representing this process: the personal best position (pBest) and the global best position (gBest). Each agent has a pBest, which is its position at a given time. Beyond this, each agent has a gBest, which is calculated using the information it procured from other agents' positions. The agent uses gBest to utilize global knowledge and find an optimal solution.

The classic PSO algorithm is shown in Equations (1) and (2). The first equation defines the agent's next step, and the second equation defines the velocity for the next step [43]. In this case, $\omega$ is the inertia weight for the velocity, and $c_1$ and $c_2$ control the local and global exploration of the agent.

$$x_i^{t+1} = x_i^t + v_i^t \tag{1}$$

$$v_i^{t+1} = \omega v_i^t + c_1 r_{1i}\left(pBest_i^t - x_i^t\right) + c_2 r_{2i}\left(gBest_i^t - x_i^t\right) \tag{2}$$

The original PSO algorithm was developed for continuous action spaces, and some PSO derivates work in discrete action spaces, such as the binary PSO. However, optimization algorithms must fit the problems they need to solve [43]. In this study, to obtain the PSO to fit the problem of the network agents detecting vulnerabilities in the network, the mathematical model of the PSO for the velocity was adjusted, which is displayed in Equation (3).

$$v_i^{t+1} = \begin{cases} gBest, & gBest > pBest \ \wedge \omega > R \begin{cases} 1 \\ 0 \end{cases} \\ pBest + \delta, & gBest < pBest \ \vee \neg \ gBest \end{cases} \tag{3}$$

Equation (3) shows the adjusted algorithm used for the PSO algorithm for this work. This adjustment affects only the velocity function. The original PSO uses a vector multiplication to calculate the agent's velocity for the next step. Therefore, the next step in the original PSO can be seen as a displacement of the agent. The adjusted PSO for this work uses both a displacement and a replacement.

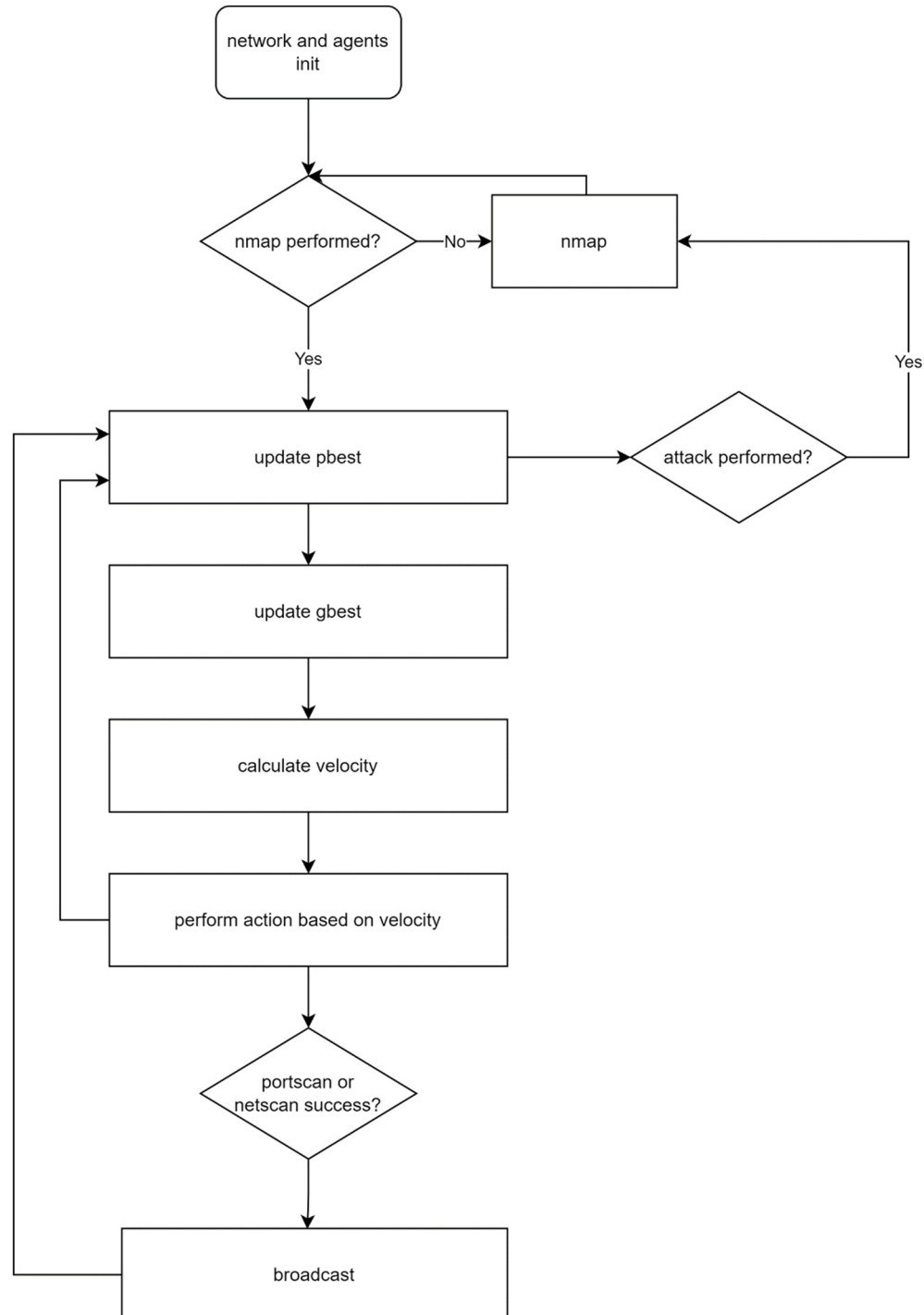

**Figure 3.** Swarm-based penetration testing algorithm utilizing PSO in CyberSim-SwarmIoT.

If the gBest in the agent's knowledge is better than the pBest, it can take this position to replace the pBest. This replacement can be controlled with the inertia weight $\omega$. The lower the inertia weight is, the lower the probability is that the agent will replace its pBest with a better gBest. When the gBest position is not used, the agent will continue in the search space with $\delta$ step, which is the step size of the next port. The inertia weight influences the exploration of the agent. The lower the inertia weight is, the higher the exploration is since the agent will continue along the search space.

In addition to the inertia weight, the communication level also affects exploration. The higher the communication level is, the higher the chance is that an agent will learn

better positions from other agents. Without knowing better positions, the agent continues exploring the state space. Therefore, communication level and inertia weight can be used to balance exploration and exploitation in the network with the PSO-based swarm algorithm.

The optimization goal of a PSO algorithm is to find an optimal solution, such as a local minimum of a function. The optimal solution for the PSO algorithm used in CyberSim-SwarmIoT is an attack action. Once an agent has performed an attack, it is dismissed and begins again with a nmap action.

### 3.5. Experiments

To compare linear penetration testing against swarm-based penetration testing, CyberSim-SwarmIoT was used to simulate networks of IoT devices. Two different scales of these networks were used, the first of which was a smart home with 30 devices, which would approximately be the number of IoT devices for a two-to-three-person household today. The second scale was a smart building with 250 devices. This was chosen for two reasons. It is an increase by multiple times of the smart home scale. Office buildings can vary in size, making different numbers of IoT devices seem feasible. Further, 250 devices and agents proved to be a number that was capable of being computed with the hardware resources available to the researchers.

To examine research objectives 1, 2, and 3, the smaller scale of the smart home was used. To examine research objective 4, the smart building scale was used.

Each simulation run on the smart home scale and smart building had a length of 50,000 timesteps, which proved to be long enough to show if the vulnerabilities in the network have been detected and provide enough insight on different algorithm dynamics. Each simulation was executed 50 times. This led to a power (1-$\beta$ error probability) of 0.7989 for a one-tailed t-test with a medium Cohen's d effect size of 0.5 and an $\alpha$ error probability of 0.05. There was a slight deviation for the smart building scale. One simulation run for the PSO algorithm failed to log results, which resulted in only 49 simulation runs usable for the analysis. This reduced the power (1-$\beta$ error probability) by 0.0036 to 0.7953.

### 3.5.1. Smart Home Scale

The network configuration for the smart home scale comprised 30 IoT devices (Table 1). Four vulnerabilities were to be detected by the agents.

**Table 1.** Network configuration for the simulated smart home environment.

| Device No. | IP | Port | Service | Vulnerability |
| --- | --- | --- | --- | --- |
| 1 | 192.168.0.1 | NONE | ping | NONE |
| 2 | 192.168.0.2 | NONE | ping | NONE |
| 3 | 192.168.0.20 | 443 | apache | sql injection |
| 3 | 192.168.0.20 | 22 | ssh | NONE |
| 3 | 192.168.0.20 | 43 | whois | NONE |
| 4 | 192.168.0.21 | NONE | ping | NONE |
| 5 | 192.168.0.22 | 22 | ssh | password crack |
| 6 | 192.168.0.23 | NONE | ssh | NONE |
| 7 | 192.168.0.24 | 80 | apache | NONE |
| 8 | 192.168.0.30 | 80 | apache | NONE |
| 9 | 192.168.0.31 | 80 | apache | NONE |
| 10 | 192.168.0.32 | 80 | apache | sql injection |
| 11 | 192.168.0.40 | NONE | ping | NONE |
| 12 | 192.168.0.60 | NONE | ping | NONE |

**Table 1.** *Cont.*

| Device No. | IP | Port | Service | Vulnerability |
|:---:|:---:|:---:|:---:|:---:|
| 13 | 192.168.0.61 | NONE | ping | NONE |
| 14 | 192.168.0.62 | NONE | ping | NONE |
| 15 | 192.168.0.63 | NONE | ping | NONE |
| 16 | 192.168.0.64 | NONE | ping | NONE |
| 17 | 192.168.0.65 | NONE | ping | NONE |
| 18 | 192.168.0.66 | NONE | ping | NONE |
| 19 | 192.168.0.67 | NONE | ping | NONE |
| 20 | 192.168.0.68 | NONE | ping | NONE |
| 21 | 192.168.0.69 | NONE | ping | NONE |
| 22 | 192.168.0.101 | 3306 | mysql | default password |
| 22 | 192.168.0.101 | 43 | whois | NONE |
| 23 | 192.168.0.102 | 43 | whois | NONE |
| 24 | 192.168.0.110 | 43 | whois | NONE |
| 25 | 192.168.0.111 | 43 | whois | NONE |
| 26 | 192.168.0.200 | NONE | ping | NONE |
| 27 | 192.168.0.201 | NONE | ping | NONE |
| 28 | 192.168.0.202 | NONE | ping | NONE |
| 29 | 192.168.0.203 | NONE | ping | NONE |
| 30 | 192.168.0.204 | NONE | ping | NONE |

For the linear algorithm (Figure 1) utilizing a single agent, the first device with IP 192.168.0.1 was defined as the penetration testing agent. This was the only agent performing penetration testing actions in this scenario. For the linear algorithm with multiple agents, each agent was using the linear penetration testing algorithm, resulting in 30 agents in the simulation.

Both swarm algorithms utilized the same network configuration. However, all devices were used as penetration testing agents. Therefore, the swarm consisted of 30 swarm agents. The adjustable parameters used for the swarm were as follows: the communication weight of the queue-based swarm algorithm was set to 3, which resulted practically in an est. 90% of the other agents reached in an agent's broadcasting period. The communication weight of the PSO algorithm was also 3.0, and the inertia weight was set to 1.0, meaning that an agent will take a gBest with a probability of 100% when this is better than the pBest.

The ports that can be used for port scans contain ports often used by devices in a network [44], meaning that 16 ports of each device were scanned instead of the entire port range. This limitation keeps the simulation simple and reduces exploration space. Table 2 lists the ports that were possible to be scanned by the agents.

The success probabilities for the penetration testing actions are listed in Table 3. A probability of 0.95 was chosen for all network actions except the password crack and nmap. This value was selected to provide sufficient action–effect possibility but take account of network traffic error to add realism. However, a further distinction of probabilities was not set to avoid creating an extraneous variable. Nmap was assigned a chance of 1.0 to always provide the algorithms with a network table at the beginning of a simulation run. The password cracking action has a very low success probability of 0.01; therefore, many password crack attacks must be performed on the device with this vulnerability until this action succeeds. This low success probability for this singular attack action was reasoned to provide insight on the effect of hard-to-detect vulnerabilities in the network, since both

scales, the smart home and the larger smart building, have vulnerabilities built in that can only be detected with the use of the password crack action.

**Table 2.** Common ports used in a network: the table is from Kulkarni, 2018; ports 3306 and 8080 were added.

| Port No. | Usage |
|:---:|:---|
| 20 | File Transfer Protocol (FTP) |
| 21 | File Transfer Protocol (FTP) |
| 22 | Secure Shell (SSH) |
| 23 | Telnet |
| 25 | Simple Mail Transfer Protocol (SMTP) |
| 53 | Domain Name System (DNS) service |
| 80 | Hypertext Transfer Protocol (HTTP) |
| 8080 | Hypertext Transfer Protocol (HTTP) |
| 110 | Post Office Protocol (POP3) |
| 119 | Network News Transfer Protocol (NNTP) |
| 123 | Network Time Protocol (NTP) |
| 143 | Internet Message Access Protocol (IMAP) |
| 161 | Simple Network Management Protocol (SNMP) |
| 194 | Internet Relay Chat (IRC) |
| 443 | HTTP Secure (HTTPS) HTTP over TLS/SSL |
| 3306 | MySQL |
| 20 | File Transfer Protocol (FTP) |
| 21 | File Transfer Protocol (FTP) |
| 22 | Secure Shell (SSH) |
| 23 | Telnet |
| 25 | Simple Mail Transfer Protocol (SMTP) |

**Table 3.** Success probabilities used for the smart home and smart building scales. These values are stored in the CyberSim-SwarmIoT look-up tables.

| Action | Success Probability |
|:---:|:---:|
| ping | 0.95 |
| port scan | 0.95 |
| netstat | 0.95 |
| sql injection | 0.95 |

Executing 50 simulation runs using the linear penetration testing algorithm with a single agent took 2 min and 56 s. With the linear penetration testing algorithm with multiple devices, this process took 1 h, 4 min, and 45 s. Next, with the queue-based swarm algorithm, it took 2 h, 44 min, and 29 s, and with the PSO swarm algorithm, it took 1 h, 57 min, and 5 s. Finally, the simulation data were analyzed with descriptive and inferential statistics. All simulation runs were performed on an AMD Ryzen Threadripper 3990X 64C running Ubuntu Linux.

### 3.5.2. Smart Building Scale

The experiment settings on the larger scale were mostly the same as those on the smart home scale. However, the following parameters were different: the smart building scale utilized a network configuration with 250 devices. There were 12 vulnerabilities to be found. The communication weight for the queue-based and the PSO-based swarm algorithms was set to 0.5, resulting in 15% of the communication range. The inertia weight for the PSO-based algorithm was 0.25, or 25%. Optimization runs demonstrated that a larger network could work with smaller communication and inertia weights. Lower inertia weights improved the exploration of the network configuration.

Executing 50 simulation runs using the linear penetration testing algorithm with a single agent took 3 min and 46 s. With the linear penetration testing algorithm with multiple

devices, this process took 2 days, 3 h, 50 min, and 38 s. Next, with the queue-based swarm algorithm, it took 4 days, 19 h, 14 min, and 59 s, and with the PSO swarm algorithm, it took 3 days, 6 h, 19 min, and 29 s. All simulation runs were performed on the same system as described for the computation of the smart home scale.

## 4. Results

The results are displayed separately for the scales of the smart home and smart building. The results for research objectives 1, 2, and 3 are recorded in the section on smart home results, and results for research objective 4 are noted in the section on the smart building results. The experiments are analyzed with descriptive and inferential statistics.

### 4.1. Smart Home Scale

On the smart home scale, four different simulations were performed utilizing three different algorithms: linear and queue-based and PSO-based. Each simulation was performed 50 times. Plotted data illuminate two different metrics: unique detected vulnerabilities and all detected vulnerabilities.

The unique vulnerabilities detected show the overall performance of the algorithms after the 50,000 timesteps. The goal of these algorithms is to detect all four vulnerabilities that can be detected in the network.

Figure 4 compares all four simulations on the smart home scale. The single-agent linear algorithm had the worst performance, with a mean of 2.58 unique vulnerabilities detected. The PSO algorithm performed best, with a mean of 3.86 unique detected vulnerabilities. The linear algorithm with multiple agents and the queue-based swarm algorithm performed similarly, with 3.14 and 3.26 unique vulnerabilities, respectively. Additionally, four vulnerabilities were detected in the network.

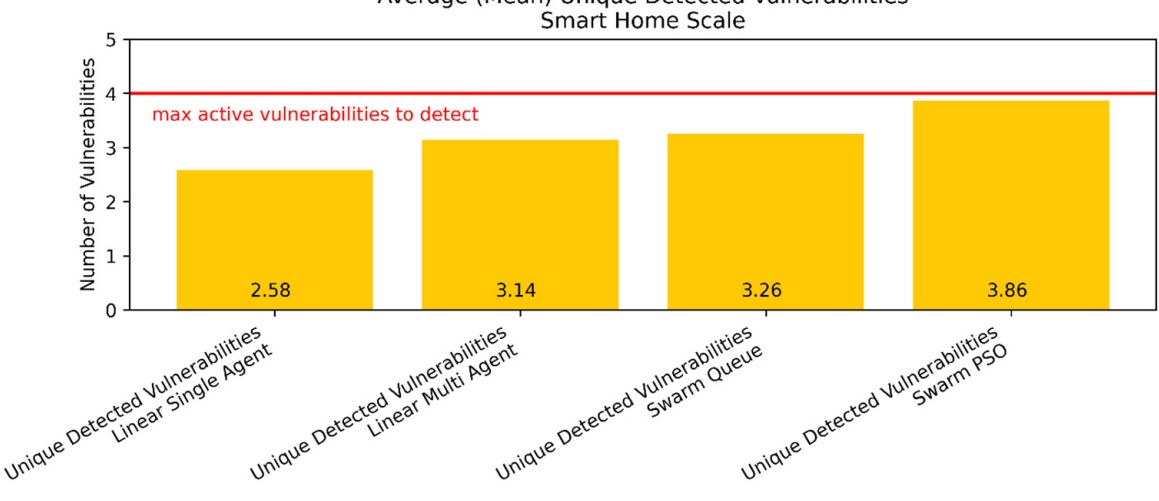

**Figure 4.** Average unique vulnerabilities detected for the smart home scale.

Figure 5 displays the unique detected vulnerabilities over time. The queue-based swarm algorithm was faster than the PSO-based swarm algorithm in detecting more than three vulnerabilities in the mean in the first 10,000 timesteps. However, the PSO algorithm produced better results over time, outperforming the queue-based algorithm at approximately 15,000 timesteps. The linear algorithm utilizing multiple agents could perform similarly to the queue-based algorithm, but this result was reached very late, after over 30,000 timesteps. The simulation with the linear single agent performed the worst of the four simulation runs.

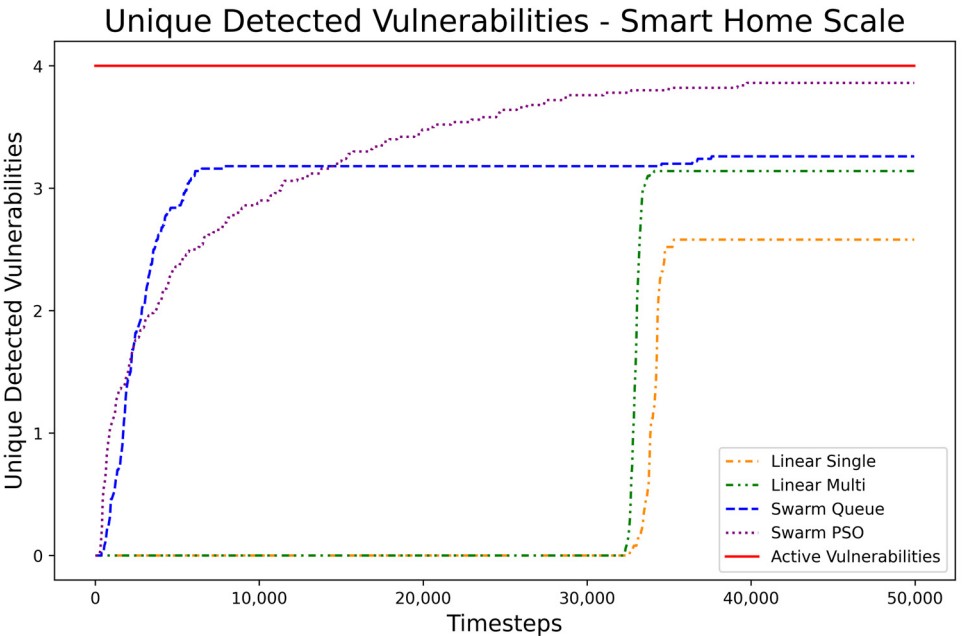

**Figure 5.** Uniquely detected vulnerabilities on the smart home scale.

To test whether the differences in detecting unique vulnerabilities were statistically significant, the researchers performed a single-factor ANOVA. The results showed significant differences between the simulations, and then Student t-tests were conducted to reveal differences between different pairs of simulations and to answer the research hypotheses.

Hypothesis 1 predicted that linear multi-agent penetration testing by other IoT devices in the same network would have a better detection rate of unique vulnerabilities than linear single-agent penetration testing. An independent one-tailed t-test was conducted to compare the linear algorithm utilizing a single agent with the linear algorithm utilizing multiple agents. The linear algorithm using a single agent (M = 2.58, SD = 0.45) compared with the linear algorithm using multiple agents (M = 3.14, SD = 0.12) demonstrated significantly better results for unique detected vulnerabilities in the time period of 50,000 timesteps (t (74) = −5.22, $p < 0.001$).

Hypothesis 2 predicted that swarm-based penetration testing by other IoT devices in the same network utilizing the queue-based algorithm would have a better detection rate of unique vulnerabilities than linear multi-agent penetration testing. An independent one-tailed t-test was conducted to compare the linear algorithm utilizing multiple agents with the queue-based swarm algorithm. There was no significant difference (t (93) = 1.66, $p = 0.068$) in the linear algorithm using multiple agents (M = 3.14, SD = 0.12) compared with the swarm-based algorithm utilizing queues (M = 3.26, SD = 0.20) for unique detected vulnerabilities in the time period of 50,000 timesteps.

Hypothesis 3 predicted that swarm-based penetration testing by other IoT devices in the same network utilizing the PSO-based algorithm would have a better detection rate of unique vulnerabilities than linear multi-agent penetration testing. An independent one-tailed t-test was conducted to compare the linear algorithm utilizing multiple agents with the PSO-based swarm algorithm. The linear algorithm using multiple agents (M = 3.14, SD = 0.12) compared with the swarm-based algorithm utilizing PSO (M = 3.86, SD = 0.12) demonstrated significantly better results for unique detected vulnerabilities in the time period of 50,000 timesteps (t (98) = 1.66, $p < 0.001$).

Therefore, among the swarm algorithms, only the PSO algorithm showed more significant results than the linear algorithm. However, both swarm algorithms detected unique vulnerabilities faster than the linear penetration testing, as demonstrated in the descriptive statistics above.

Hypothesis 4 predicted that swarm-based penetration testing by other IoT devices in the same network utilizing the PSO-based algorithm would have a better detection rate of unique vulnerabilities than swarm-based penetration testing utilizing the queue-based algorithm. An independent one-tailed t-test was conducted to compare the queue-based swarm algorithm and the PSO-based swarm algorithm. The queue-based swarm algorithm (M = 3.26, SD = 0.20) compared with the linear algorithm using multiple agents (M = 3.86, SD = 0.12) demonstrated significantly better results for unique detected vulnerabilities in the period of 50,000 timesteps (t (93) = −7.51, $p < 0.001$).

Figure 6 displays all the detected vulnerabilities, including multiple detections of vulnerabilities. Notably, the single linear algorithm, utilizing only a single agent, provides all detected vulnerabilities at the same level as successful unique detected vulnerabilities (Figure 5). When applied by multiple agents, it can be seen how vulnerabilities are detected multiple times. The swarm-based algorithms showed numerous multiple vulnerabilities detected very early in the simulation. In comparison to the PSO, the queue-based algorithm shows increases that lead to plateaus. A possible explanation for that is given in the Discussion section.

**Figure 6.** All detected vulnerabilities on smart home scale.

### 4.2. Smart Building Scale

On the smart building scale, four different simulations were performed utilizing three different algorithms: linear, queue-based, and PSO-based. Each simulation was performed 50 times. The plotted data provide insight for two different metrics: unique detected vulnerabilities and all detected vulnerabilities.

Figure 7 compares the unique vulnerabilities detected at the smart building scale. The algorithm based on PSO demonstrated the best performance, with a mean of 11.94 unique vulnerabilities detected. The queue-based algorithm followed, with a mean of 11.78 unique vulnerabilities detected. The linear algorithm provides no successful unique detected vulnerability at all. This applies to use with a single agent as well as with multiple agents. There was a maximum of 12 vulnerabilities in the network.

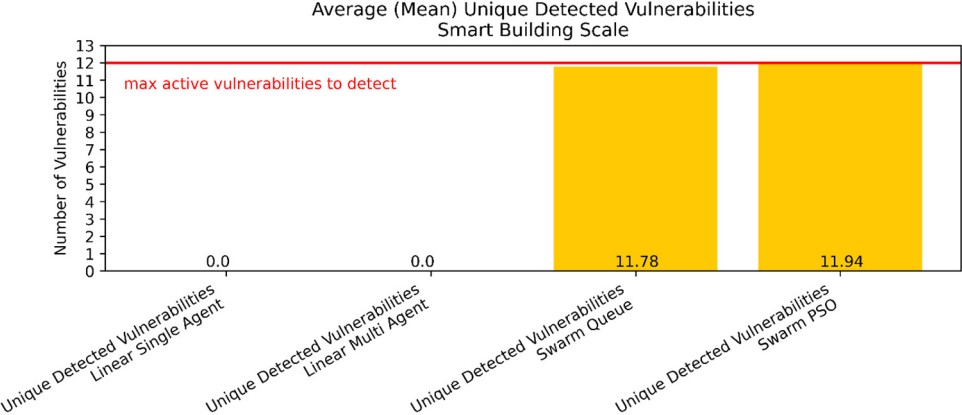

**Figure 7.** Average unique vulnerabilities detected for the smart home scale.

An independent one-tailed t-test was conducted to test both swarm algorithms for statistically significant differences. The PSO algorithm (M = 11.94, SD = 0.06) compared with the queue-based algorithm (M = 11.78, SD = 0.22) demonstrated significantly better results for unique detected vulnerabilities in the time period of 50,000 timesteps (t (73) = −2.14, *p* = 0.018).

The time plot for the unique vulnerabilities detected (Figure 8) demonstrates that the linear algorithm was not capable of detecting any vulnerability. This applies for the single agent as well as for the multi-agent experiments. For the swarm algorithms, which are the focus of research objective 4, the plot provides an equal start for the queue-based and the PSO algorithms. At 2000 timesteps, the queue algorithm starts to outperform the PSO algorithm.

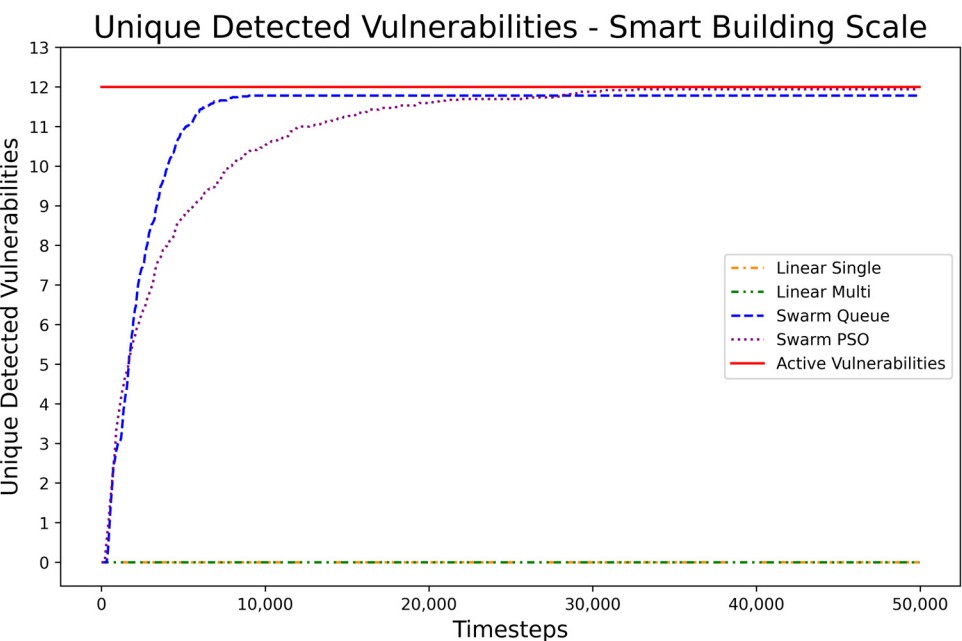

**Figure 8.** Uniquely detected vulnerabilities on smart building scale.

The time plots for all detected vulnerabilities (Figure 9) display that the queue-based algorithm reached a higher number of all detected vulnerabilities very early in the period of 50,000 timesteps. It took the PSO-based algorithm more than 40,000 timesteps to reach to the level of the queue-based algorithm. The linear algorithm remained with no vulnerabilities detected, for both the single agent as well as the multi-agent approach.

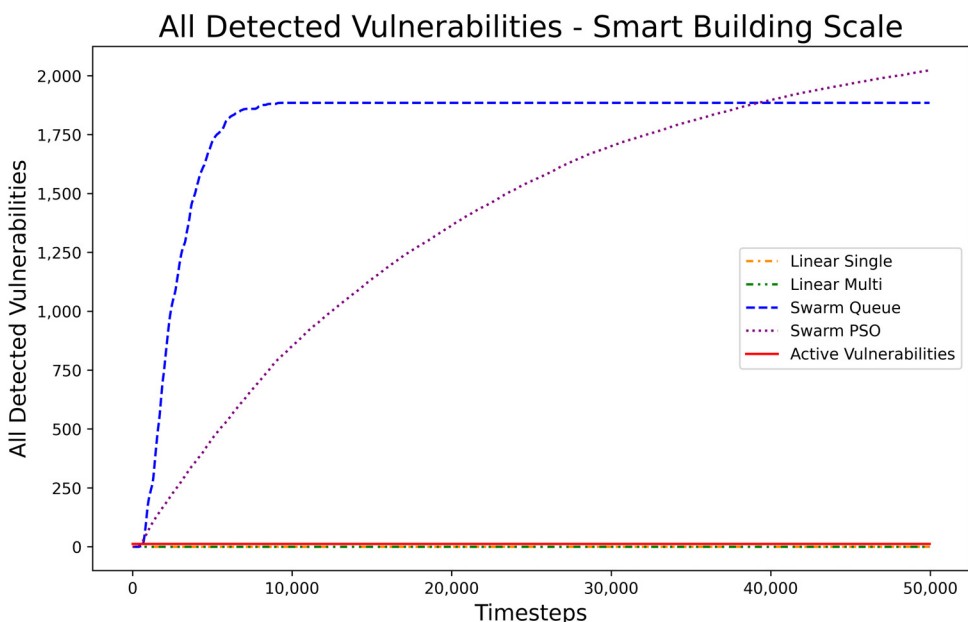

**Figure 9.** All detected vulnerabilities on the smart building scale.

## 5. Discussion

All three algorithms used in these experiments yielded unique performance differences and dynamics to the penetration testing problem. Thus, this discussion describes the differences in the results, explaining the specific characteristics of the linear and both swarm-based algorithms. This section closes with limitations that appeared during this work.

In this study, the linear algorithm detected vulnerabilities late because of the algorithm's architecture, which provides the chronology of scanning, enumerating, and attacking. Therefore, the scanning and enumerating take longer with more ports or devices to be scanned, which explains why the agents using the linear algorithm could detect vulnerabilities on the smart home scale but not on the smart building scale. The increase in devices in the network significantly increased the ports that needed to be scanned. The linear algorithm could not finish scanning within the allotted 50,000 timesteps. When multiple devices used the linear algorithm, hard-to-detect vulnerabilities in the smart home network had a higher probability of detection because there were more devices testing in parallel. However, the issue with the late attack on the smart building scale remained. Overall, the simple structure of the algorithm produced a low computational time, and the linear algorithm was the fastest of all algorithms to compute.

The queue-based swarm algorithm performed much like the linear algorithm when applied by multiple agents over the period of 50,000 timesteps. However, detecting the vulnerabilities began earlier than with the linear algorithm and took a significant amount of time. The architecture of the algorithm ensures that, when not in broadcasting mode, the algorithm always focuses on high-priority actions because the current agent action is always based on the highest-priority queue that is not empty. Because queues are filled according to both their own information and information from other agents, they can be constantly filled with new information. Therefore, agents who use the queue algorithms can profit from a constant stream of information useful for service scanning or attacking. Yet, computing this continuous information stream and storing it in the queues has a cost. For instance, the queue-based swarm algorithm was the most time consuming to compute. Additionally, during the computation, the computation required more memory than the PSO and the linear algorithms (approximately three to four times as much). Ultimately, the queue algorithm could detect vulnerabilities faster than the PSO algorithm, but the latter

outperformed the former over time, which can be explained through the characteristics of the PSO algorithm.

The PSO-based swarm algorithm provided superior detection of vulnerabilities to the linear algorithm when used with multiple agents. On the other hand, the former was not as fast at detecting vulnerabilities as the queue-based algorithm. However, the PSO was approximately 30% faster to compute, took approximately 2–3 times less memory than the queue-based algorithm, and outperformed the queue-based algorithm over the time of 50,000 timesteps. Figures 3 and 5 of all detected vulnerabilities demonstrate that the PSO's actions were more linear over time than the queue due to the architecture and character of the PSO. The PSO algorithm only stores two chunks of information: the own best position (pBest) and the global best position (gBest). This little information stored explains why 2–3 times less memory was used for the PSO. Once an agent with the PSO algorithm reached its local minimum, defined as an attack action, this agent was reset and started over with an nmap action. This phenomenon can be represented as an agent in a computer game deleted from the map that respawns at another point on the same map. Therefore, the PSO algorithm does not contain old information on which to base its next action. However, the queue-based algorithm can contain old information in the queue that must be exhausted before the agents began again with an nmap action and a new network table to start again with a new session of penetration testing.

Given the different characteristics of the queue-based and PSO algorithms, the PSO could detect new devices with vulnerabilities in the network more quickly than the queue-based algorithm due to the higher nmap frequency of the PSO after reaching a local minimum. However, this behavior was not observed in this work since no dynamic networks were used. Furthermore, the PSO algorithm uses fewer resources because it is faster to compute and uses less memory. However, no metrics were collected in this work to test this hypothesis.

## 6. Limitations

To reduce simulation complexity and exploration space, several concessions were made compared to the full range of penetration testing parameters. For example, the port range was reduced by the researchers to reduce model testing runtime. Beyond this, increasing port numbers increased the exploration space exponentially. However, scanning all 65,535 ports yielded no meaningful results because the port scan took too much time, partially due to the current constraint in CyberSim-SwarmIoT that each agent can only scan one port at a time.

The values used for action probabilities (Table 3) were set equally to avoid having extraneous variables creating significant differences between different actions. Only the nmap action was set to full success probability to prevent the extraneous variable of agents not obtaining a network table to start the penetration testing with. The password crack action was set to a very low probability (0.01) so the researchers could learn how different algorithms handle hard-to-detect vulnerabilities. However, these values are limited in how they reflect reality. This limitation seems feasible for the simplified penetration testing simulation used in this work, but it should be investigated when using a more extended and realistic action table.

The queue-based swarm algorithm reliably detected vulnerabilities in the swarm with 3.26 out of 4 and 11.78 out of 12 vulnerabilities. However, in this process, agents sometimes stopped performing actions even though elements were available in the queues. This implementation problem could not be solved entirely during this work. A partial solution was used to "wake up" agents after a certain period of inactivity and let the agents perform actions according to the queue again. Therefore, with a solution to this problem, the queue-based algorithm might perform better.

In this study, the experiments used only a static network configuration, which reduced realism but enhanced the comparison of the algorithms on a baseline level.

## 7. Conclusions

The results demonstrated that multi-agent and swarm-based penetration testing can detect vulnerabilities in an IoT network faster than traditional single-agent penetration testing. The swarm-based algorithms facilitated cohesive, synergistic penetration testing, which drastically increased vulnerability detection. The results from the larger scale revealed that the linear approach of scanning, enumerating, and exploiting failed because too much time passed until the exploitation of the possible vulnerabilities began. The swarm algorithms were successful on a larger scale due to their direct exploitation of possible vulnerabilities and their task allocation. Using multiple agents and inter-agent communication produced the possibility of sharing information about the exploration space. The non-nature-based swarm algorithm utilizing queues detected vulnerabilities faster than the nature-based PSO algorithm, but the PSO algorithm demonstrated a better detection rate of vulnerabilities over time.

Overall, swarm-based algorithms can be deployed for autonomous penetration testing in a network. The application of the base idea of using IoT devices in the network to test other IoT devices to produce safer smart homes and IoT networks could increase awareness of possible security vulnerabilities or old and unsafe devices in the network before these vulnerabilities are exploited. Additionally, existing resources that are often sparsely used in their lifetimes can be deployed for detecting penetration vulnerabilities. Over time, using these existing IoT devices could reduce the resources used while accommodating new use cases.

## 8. Future Research

This work demonstrated the simulation of swarm-based penetration testing of IoT devices by IoT devices on a base level. However, several elements could be investigated in future research.

This study used a PSO algorithm, one of the first nature-based algorithms. In this vein, many other nature-based swarm algorithms have been developed in recent decades [45,46]. Examples include the Artificial Bee Colony (ABC) algorithm, which mimics the behavior of honey bees seeking food [47]. The Gray Wolf Optimization (GWO) algorithm mimics the behavior of gray wolves, which live in a hierarchy and utilize this order when hunting prey [48]. Similarly, the grasshopper algorithm mimics the global and local search of grasshoppers seeking food [49]. These three nature-based algorithms are only a fraction of the swarm-based algorithms available, and each has its own mathematical model and optimization method. Thus, future research could implement such algorithms in CyberSim-SwarmIoT and compare the effectiveness and efficiency of the algorithms in detecting vulnerabilities in the simulated network.

Because this study used only static network configurations, future work could address this aspect with dynamic network configurations, adding realism to experiments since new devices could enter the network, and other devices could leave the network. An example of this process would be having a smartphone connect to a network when a person returns home.

In this work, the upscaling from the smart home to the smart building scale revealed that the communication level and inertia weights could be lowered because having more agents increased attacks on the same target. The two scales in this work (30 devices and 250) did not clarify the amount of reduction and the dynamic behind this parameter change. Consequently, future work could test several different network sizes and parameter settings to develop a formula for defining optimal parameters for a given network size.

Realism could be improved in future work with adding vulnerabilities according to the Common Vulnerabilities and Exposures (CVE) reference system [50]

This study used swarm-based algorithms programmed to perform in a certain way, but no ML was used to create or train the agents' behavior. Thus, future studies could utilize ML, especially RL, to optimize swarm behavior and increase the effectiveness and efficiency of vulnerability detection [51].

Furthermore, this work employed a network simulator to test swarm-based algorithms. However, simulating real-world aspects involves certain constraints since aspects must be simplified for the simulation. Therefore, the next step would be to emulate networks of IoT devices [52], which could involve virtual machines or microcomputers like a cluster of Raspberry Pis [17]. Each virtual machine or microcomputer could represent an IoT device and performs penetration testing actions based on swarm-based algorithms. Thus, such studies could better represent realistic scenarios and applications.

Networks of self-penetration-testing IoT devices include more than technical aspects. However, it is worth investigating how such an implementation would work, including aspects beyond the technology, such as potential legal issues and additional costs. These issues must be considered when implementing swarm-based penetration testing of IoT devices in a business case.

This work examined a smart home and a smart office building as a setting for swarm-based penetration testing. However, these are not the only settings where IoT devices are used. For instance, industrial settings within Industry 4.0 are highly dependent on IoT devices, which is called the Industrial Internet of Things (IIoT) [4,53]. A similar shift occurs in the military, where the terms Internet of Military Things (IoMT) and Internet of Battle Things (IoBT) describe scenarios where increasing numbers of devices and sensors on the battlefield are connected and communicate with each other [5]. Industrial or military settings have different foci that future work could examine. For instance, those in industrial settings could implement swarm-based penetration testing with zero delays, while those using military applications should consider communication aspects in the swarm, since communication might need to be reduced in this context. Using swarm-based IoT penetration testing in these settings would enhance security beyond that of smart homes and smart buildings.

**Author Contributions:** Conceptualization, T.S., B.C., A.S.W. and S.M.; methodology, T.S., B.C., A.S.W. and S.M.; software, T.S. and S.M.; validation, T.S. and S.M.; formal analysis, T.S. and S.M.; investigation, T.S.; resources, T.S.; data curation, T.S.; writing—original draft preparation, T.S.; writing—review and editing, T.S., B.C., A.S.W. and S.M.; visualization, T.S.; supervision, S.M.; project administration, S.M. All authors have read and agreed to the published version of the manuscript.

**Funding:** This research received no external funding.

**Data Availability Statement:** Data acquired in the experiments and used for analysis and results can be downloaded under the following URL: https://github.com/ThomasUCF/IoTPenTestingByIoT_Experiment_Data (accessed on 25 September 2023).

**Acknowledgments:** The authors want to thank Malic Dekkar, for constant critique and review. Additionally, the authors thank Thane Keller.

**Conflicts of Interest:** The authors have no conflict of interest.

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
