# Peer review of "Security Awareness in Smart Homes and Internet of Things Networks through Swarm-Based Cybersecurity Penetration Testing"

_information, doi:10.3390/info14100536_

Round 1

Reviewer 1 Report

This paper is targeting an optimization-based approach for penetration testing using reinforcement learning.

- The authors mention CAN in line 57 without definition.

- The authors have a dedicated section on cybersecurity measurement without any mention of well known method for this purpose, which is attack graphs. An appropriate reference, such as [1]

- line 128, "this thesis". What thesis are your referring to here?

- Please cite paper [2] regarding RL-based pentesting.

- It would be nice and beneficial to the reader if you explain a bit about Deep-Q network.

- In Figure A3, what will happen if the attack isn't performed?

- Line 326, the authors mention equation 4, but it does not exist in the paper.

- In the experiments, the author mention 4 vulnerabilities. Where do the vulnerabilities come from? Are they simulated? 

- Is your experiment environment simulated or hardware-based?

[1] Sabur, Abdulhakim, et al. "Toward scalable graph-based security analysis for cloud networks." Computer Networks 206 (2022): 108795.

[2] Chowdhary, Ankur, et al. "Autonomous security analysis and penetration testing." 2020 16th International Conference on Mobility, Sensing and Networking (MSN). IEEE, 2020.

line 157

line 319

Author Response

Dear Reviewer,

Thank you very much for your review. We have addressed your comments on our article and have proofread the paper again several times. We think that your comments helped a lot to create a better article.

In the following you can see, how we addressed your comments:

Reviewers Comment

How is the comment addressed in the revised version

The authors mention CAN in line 57 without definition.

Abbreviation is now written-out.

The authors have a dedicated section on cybersecurity measurement without any mention of well known method for this purpose, which is attack graphs. An appropriate reference, such as [1]

[1] Sabur, Abdulhakim, et al. "Toward scalable graph-based security analysis for cloud networks." Computer Networks 206 (2022): 108795.

Attack graphs are mentioned in the beginning of the section “2.3 Autonomous Penetration Testing”. The paper mentioned by the reviewer is added to this section as well.

line 128, "this thesis". What thesis are your referring to here?

It refers to this study. Changed to “study”.

Please cite paper [2] regarding RL-based pentesting.

[2] Chowdhary, Ankur, et al. "Autonomous security analysis and penetration testing." 2020 16th International Conference on Mobility, Sensing and Networking (MSN). IEEE, 2020.

Paper is now cited in the section “2.3 Autonomous Penetration Testing”.

It would be nice and beneficial to the reader if you explain a bit about Deep-Q network.

Authors opted to not explain Deep-Q Networks deeper, since this work focus on swarm as Artificial Intelligence method. Q-Networks are mentioned in relevant work.

In Figure A3, what will happen if the attack isn't performed?

Attack might be performed, when pbest is high enough. When attack is performed, the agent will go back to nmap and is therefore “respawned”.

Line 326, the authors mention equation 4, but it does not exist in the paper.

Equation 4 was used in a prior version of the draft. The mentioning of equation 4 is now deleted.

In the experiments, the author mention 4 vulnerabilities. Where do the vulnerabilities come from? Are they simulated? 

The vulnerabilities are present in the network table. But these vulnerabilities are not known to the agents. The simulation environment however, checks the blackboard of the agent against this pre-configured network table with the set of vulnerability.

Is your experiment environment simulated or hardware-based?

The experiment is simulation-based. Hardware-based solution is mentioned as future work (emulation).

Sincerely Yours,

The Authors

Reviewer 2 Report

The authors present an autonomous and swarm-based cybersecurity penetration testing platform for Internet of Things (IoT) environments. The overall work is interesting and the proposed method is explained well. Moreover, the simulation results show the efficiency of the proposed method. However, there is room for improvements.

1. The paper is not structured very well. For instance, a new section is needed, describing relevant works in this field.

2. Regarding the description of relevant works, the authors could consider comprehensive security frameworks such as SDN-microSENSE and SPEAR, identifying the contributions of the proposed solution.

3. The contributions should be stated in clear manner in the introductory part.

4. The evaluation results should be enhanced with more methods and datasets if possible.

5. The paper should be re-checked for potential typos and writing errors.

The paper should be re-checked for potential typos and writing errors.

Author Response

Dear Reviewer,

Thank you very much for your review. We have addressed your comments on our article and have proofread the paper again several times. We think that your comments helped a lot to create a better article.

In the following you can see, how we addressed your comments:

Reviewers Comment

How is the comment addressed in the revised version

The paper is not structured very well. For instance, a new section is needed, describing relevant works in this field.

Background section is now changed to relevant work, to make clear, that prior literature in the field is handled here. Further, the section has an introductory paragraph to provide detailed information to the reader what will be displayed.

Regarding the description of relevant works, the authors could consider comprehensive security frameworks such as SDN-microSENSE and SPEAR, identifying the contributions of the proposed solution.

SDN-microSENSE and SPEAR SIEM are added to the section “2. Related Work” at the end of the subsection “2.2. Cybersecurity measures”.

The contributions should be stated in clear manner in the introductory part.

Contributions of this work are now clearly stated at the end of the introduction chapter.

The evaluation results should be enhanced with more methods and datasets if possible.

Complete datasets of this work (10 GB uncompressed) is available at GitHub. Including Jupyter Notebook Files used for the analysis. https://github.com/ThomasUCF/
IoTPenTestingByIoT_Experiment_Data

This GitHub is mentioned at the end of the article.

The paper should be re-checked for potential typos and writing errors.

Paper is rechecked for typos and writing errors.

Sincerely Yours,

The Authors

Reviewer 3 Report

Summary: "This work proposed a new method that makes IoT networks safer using autonomous and swarm-based cybersecurity penetration testing. Three different algorithms (linear, queue-based, and particle-swarm optimization [PSO]) are used to find vulnerabilities in the network."

Major Comments:

  1. The introduction part is shorter. The authors should cite some more recent papers focusing on IoT Security. 
    1. You, I., Kwon, S., Choudhary, G., Sharma, V. and Seo, J.T., 2018. An enhanced LoRaWAN security protocol for privacy preservation in IoT with a case study on a smart factory-enabled parking system. Sensors18(6), p.1888.                                               
    2. Kachare, G.P., Choudhary, G., Shandilya, S.K. and Sihag, V., 2022, February. Sandbox Environment for Real-Time Malware Analysis of IoT Devices. In International Conference on Computing Science, Communication and Security (pp. 169-183). Cham: Springer International Publishing
  2. Authors should explicitly specify the novelty of the proposed work. What progress against the most recent state-of-the-art similar solution was made?
  3. The problem statement and key contribution are missing. The authors should add a problem statement and key contribution subsection in the introduction.
  4. The proposed approach should be compared with the existing method to verify and justify the accuracy and other factors.
  5. Tables and flow charts should be added to the paper instead of the appendix and referred to in the text.

 English throughout the manuscript needs to be improved.

Author Response

Dear Reviewer,

Thank you very much for your review. We have addressed your comments on our article and have proofread the paper again several times. We think that your comments helped a lot to create a better article.

In the following you can see, how we addressed your comments:

Reviewers Comment

How is the comment addressed in the revised version

The introduction part is shorter. The authors should cite some more recent papers focusing on IoT Security. 

1.     You, I., Kwon, S., Choudhary, G., Sharma, V. and Seo, J.T., 2018. An enhanced LoRaWAN security protocol for privacy preservation in IoT with a case study on a smart factory-enabled parking system. Sensors18(6), p.1888.

2.     Kachare, G.P., Choudhary, G., Shandilya, S.K. and Sihag, V., 2022, February. Sandbox Environment for Real-Time Malware Analysis of IoT Devices. In International Conference on Computing Science, Communication and Security (pp. 169-183). Cham: Springer International Publishing

Both papers mentioned by the reviewer are not cited in the section “2. Related Work”.

Authors should explicitly specify the novelty of the proposed work. What progress against the most recent state-of-the-art similar solution was made?

Novelty is now explicitly mentioned in Introduction section.

The problem statement and key contribution are missing. The authors should add a problem statement and key contribution subsection in the introduction.

Problem statement and key contribution are add to the Introduction section.

The proposed approach should be compared with the existing method to verify and justify the accuracy and other factors.

The existing method is simulated with the linear algorithm. Both swarm algorithms (queue-based and PSO-based) are compared against the linear approach.

Tables and flow charts should be added to the paper instead of the appendix and referred to in the text.

Tables and flowchart are now added to the paper and moved out of the appendixes.

Sincerely Yours,

The Authors

Round 2

Reviewer 1 Report

Thank you for answering my previous questions and comments.

However, the authors seem not to understand my concern regarding the vulnerabilities. It's mentioned that the vulnerabilities are presented in the network table. But I don't see any info in the regard in Table 2. I was expecting to see a CVE ID, CVSS score, and above all, an explanation on where do the vulnerabilities come from? Are they planted by you? Are they embedded in the simulation environment? 

Please have the manuscript revised one more time as I see some minor mistakes. 

Author Response

Dear Reviewer,   Thank you very much again for your comments in the review round 2.   The vulnerabilities are "hardcoded" in the network table. Therefore they are planted by the authors. The vulnerabilities are examples of what the approach can detect. In the future, these example vulnerabilities can be replaced by others, including ones identified by NIST or common IoT vulnerability lists.   Your comment on the CVE reference system is very good and using the CVE reference system would definitely improve the work's quality and would add more realism. Therefore, we have added the following paragraph to the future research section:   "Realism could be improved in future work with adding vulnerabilities according to the Common Vulnerabilities and Exposures (CVE) reference system [50]"   [50] P. Mell and T. Grance, “Use of the Common Vulnerabilities and Exposures (CVE) vulnerability naming scheme,” National Institute of Standards and Technology, Gaithersburg, MD, NIST SP 800-51, 2002. doi: 10.6028/NIST.SP.800-51.   Further we did more proofreading and were able to detect and diminish more grammatical errors and hopefully improve the language of the paper to your expectation.   We would like to thank you again for your comment and hope, that we improved the work according to your expectations.   Sincerely, The Authors

Reviewer 2 Report

The authors addressed the relevant comments; therefore, the paper can be accepted

Author Response

Dear Reviewer,

Thank you very much for your comment and reviewing work. We are looking forward to publish this paper.

Sincerely,

The Authors

Reviewer 3 Report

The authors have significantly improved the article. There are no further comments. 

Author Response

(The authors gave the same response as above.)
